# Similarity-Based Virtual Screening to Find Antituberculosis Agents Based on Novel Scaffolds: Design, Syntheses and Pharmacological Assays

**DOI:** 10.3390/ijms232315057

**Published:** 2022-12-01

**Authors:** Ángela García-García, Jesus Vicente de Julián-Ortiz, Jorge Gálvez, David Font, Carles Ayats, María del Remedio Guna Serrano, Carlos Muñoz-Collado, Rafael Borrás, José Manuel Villalgordo

**Affiliations:** 1Unidad de Investigación de Diseño de Fármacos y Conectividad Molecular, Departamento de Química Física, Facultad de Farmacia, Universitat de València, 46100 Burjassot, Spain; 2Departamento de Química, Universitat de Girona, 17071 Girona, Spain; 3Departamento de Microbiología, Facultad de Medicina y Odontología, Universitat de València, 46010 València, Spain

**Keywords:** MTBC, virtual screening, topological indices, linear discriminant analysis, pharmacological activity distribution diagrams, antimicrobial drugs, drug design

## Abstract

A method to identify molecular scaffolds potentially active against the Mycobacterium tuberculosis complex (MTBC) is developed. A set of structurally heterogeneous agents against MTBC was used to obtain a mathematical model based on topological descriptors. This model was statistically validated through a Leave-n-Out test. It successfully discriminated between active or inactive compounds over 86% in database sets. It was also useful to select new potential antituberculosis compounds in external databases. The selection of new substituted pyrimidines, pyrimidones and triazolo[1,5-*a*]pyrimidines was particularly interesting because these structures could provide new scaffolds in this field. The seven selected candidates were synthesized and six of them showed activity in vitro.

## 1. Introduction

Tuberculosis is one of the deadliest infections in the world, killing almost 1.45 million people annually [1,2]. Most of these deaths occur in poor countries, and a more rational distribution of wealth could prevent them. However, the increasing incidence of drug-resistant MTBC has caused its resurgence in developed countries and makes therapeutic alternatives necessary [3,4]. Although new compounds potentially active against these bacteria are under active research, very few new treatments have been developed in recent years [5,6,7,8,9]. Therefore, more extensive investigations are needed to find new molecular scaffolds capable of generating new inhibitors of mycobacteria.

The α-acetylenic ketones of type A (Figure 1) have been shown to be highly versatile building blocks. For instance, these conjugated ynones have proven to be very suitable substrates for the synthesis of (E)-3-acylpropenoic acids [10], and of a wide range of heterocyclic systems [11,12,13,14,15,16,17,18,19,20,21], including the synthesis of the natural product L-lathyrine and related analogues [22,23]. Furthermore, when properly functionalized, compounds of type A have also proven to be valuable substrates for combinatorial and parallel synthesis on solid support of highly molecular diverse 2,4,6-trisubstituted pyrimidines [24,25,26,27]. These α-acetylenic ketones are produced readily, mainly by direct palladium-catalyzed coupling of acyl chlorides with 1-alkynes (Sonogashira reaction), [28,29] by reaction of alkynylzinc chlorides with acid halides, [30,31] or by reaction of lithium or magnesium acetylides with aldehydes followed by subsequent oxidation with oxalyl chloride in DMSO (Swern oxidation) [32], MnO_2_ [12,13,15,33], t-butyl hydroperoxide [34] or with IBX [33,34,35].

In this paper, we have further expanded the synthetic utility of these conjugated ynones A, and we wish to report the synthesis of novel 2,5,7-substituted triazolo[1,5-*a*]pyrimidines B by cyclocondensation of different α-acetylenic ketones A with 3-amino-5-benzylsulfanyl-1,2,4-triazole (Figure 1).

The triazolo[1,5-*a*]pyrimidine nucleus is of considerable chemical and pharmacological interest. Antibiotic, anticholesteremic, antidiabetic, antiallergic, anti-inflammatory, antipyretic, antiphlogistic, analgesic and anticancer activities have been described for these types of compounds. They also serve in the treatment and prevention of circulatory diseases such as hypertension, heart diseases, stroke, hypercholesterol, arteriosclerosis and are effective coronary vasodilators and bronchodilators [36,37,38,39,40,41,42,43,44,45,46,47,48,49].

On the other hand, several triazolo[1,5-*a*]pyrimidine derivatives have found applications in the agrochemical industry. Additionally, 1,2,4-triazolo[1,5-*a*]pyrimidinesulfonamides are used as herbicides and plant growth inhibitors, and they show activity against acetolactate synthase [46,50,51].

Currently, there are many methods available to synthesize triazolo[1,5-*a*]pyrimidines, mainly they are cyclocondensations between 3-amino-1,2,4-triazoles and 1,3-bifunctional synthons such as 1,3-dicarbonyl compounds or their equivalents [52,53,54,55,56,57,58,59], vinylogous iminium salts [47,60,61], ketene dithioacetals [48,62,63], and 3-ketovinyl compounds [46]. We used α-acetylenic Ketones of types A as 1,3-bifunctional synthons.

Although a few pyrimidines [64], pyrimidones [65] and triazolo[1,5-*a*]pyrimidines [66] have already been synthetized and tested for anti-TB activity, their substitution patterns and synthesis methods are different from the compounds described here.

The identification of new targets requires knowledge of the specific biochemical pathways of mycobacteria, but many metabolic processes are still unknown and the structure-based design of new anti-TB agents is a complex task [67].

On the other hand, extra-mechanistic virtual screening methodologies have demonstrated their ability to model the presence of activity within structurally heterogeneous groups of compounds in different therapeutic areas [68,69,70,71,72,73,74,75,76,77] as well as in predicting toxicological properties [78,79] and drug-like characteristics [80]. In these models, structural similarity is the key. Molecules are characterized through structural invariants, that is, by descriptors that are independent of molecular conformation. Many of them are topological indices (TI) [81,82,83,84,85,86,87], which are capable of characterizing most of the molecular structure [88,89,90,91,92,93,94,95].

The aim of this study was to develop new discriminant models based on the molecular structures of antituberculosis agents, shown in Appendix A [96,97,98] and inactive substances, shown in Appendix A, characterized by topological indices [99] (Table 1), in order to screen structural databases to identify new potentially useful scaffolds against MTBC.

## 2. Results and Discussion

### 2.1. Antituberculosis Activity Modelling

Figure 1 illustrates the proposed virtual screening procedure. In this method, thresholds on descriptor values were combined with Linear Discriminant Analysis (LDA) to give a qualitative discriminant model. Appendix A collects all these values.

Given a population, for example of molecules, that can be classified into several groups according to their experimental properties, for example a group of molecules with a pharmacological activity and another without this activity, LDA is a method to find linear combinations of independent variables (for example the aforementioned structural invariants) whose numerical values can be used to distinguish between these different categories. When two categories are defined, the classification is done by the so-called discriminant function (DF).

The following equation DF was obtained by stepwise LDA using the set shown in Table 2:DF = 0.88 − 11.99 J_1_ + 4.86 J_1_^v^ − 11.11 J_3_^v^ + 0.81 ^1^D − 0.2 ^4^C_c_
N = 60 λ = 0.34 F = 21.12
where N = 60 represents 25 anti-MTBC drugs and 35 presumably inactive compounds.

The topological descriptors selected in this equation were: the charge indices (J_1_, J_1_^v^, J_3_^v^); the difference index (^1^D = ^1^χ − ^1^χ^v^); and the quotient index (^4^C_c_ = ^4^χ_c_/^4^χ_c_^v^), where ^m^χ_t_ and ^m^χ_t_^v^ are, respectively, single and valence Randić-Kier-Hall indices of order m and type t. Topological charge-transfer indices, J_1_, J_1_^v^ and J_3_^v^, are measures of the contribution of molecular topological structure to the charge transfer at topological distance 1 and 3, respectively [68,101]. Difference and quotient indices are related to charge distributions within molecular fragments [68]. Thus, ^1^D is the net contribution of the heteroatoms to the electronic charge within fragments of order 1 (bonds). The ^4^C_c_ index can be related to electron densities of cluster-type fragments of order 4 (three atoms bonded to a central one) in which there is at least one heteroatom. The maximum correlation between pairs of these selected variables, for the group of actives, was as weak as 0.424, corresponding to the pair J_1_^v^ and J_3_^v^. Thus, the variables can be considered as independent. Appendix A shows these intercorrelations.

Table 2 shows the results of the classification for each one of the compounds included in the LDA.

The linear equation gave good results since most compounds were classified with a probability over 86% (Table 2). When the probabilities are between 40% and 60%, the compounds were counted as non-classified (NC), and finally below 40%, they were considered as inactive. Under this framework, the error percentage in the active set was about 9%, whereas in the inactive was 0%. These probabilities are the so-called posterior probabilities, computed by the Bayes rule as the probability of classifying a case (molecule) conditioned to the model obtained. Let π_k_ the prior probability: $\pi_{k}=\frac{Nº\;of\;cases\;in\;class\;k}{Total\;Nº\;of\;cases}$. The posterior probability *P* is given by: $P=\frac{f_{k}\left(x\right)\pi_{k}}{\sum_{i=1}^{k}f_{i}\left(x\right)\pi_{i}}$ where $f_{k}\left(x\right)$ is the class-conditional density of the case *x* in the class *k*. Assuming that this density for x, given every class k, follows a normal distribution, the density formula for a multivariate Gaussian distribution is applicable. Thus, $f_{k}\left(x\right)=\frac{\exp\left[-\frac{1}{2}{\left(x-\mu_{k}\right)}^{T}Cov_{k}^{-1}\left(x-\mu_{k}\right)\right]}{\sqrt{{\left(2\pi \right)}^{p}\cdot \det\left(Cov\right)}}$ where x and the mean $\mu_{k}$ are both column vectors, *Cov* is the covariance matrix and *p* is its dimension. The denominator involves the square root of the determinant of this matrix. The result of the matrix multiplications in the numerator is a scalar number.

The results of the internal validation are illustrated in Table 3, with the percentage of correct classification within each group. Five runs were performed. A number of compounds ranging from 9 to 16 were randomly extracted from the training to a test set. Wilks λ values are shown for each equation. The lower the Wilks λ value, the better the discrimination. Correct classification percentages are shown for training and test sets, for active and inactive compounds. The number of compounds classified as active (+) or inactive (−) appears in parentheses, where (a/b) = number of (+) compounds/number of (−) compounds. Average values are also shown, as well as the performance of DF function.

The results for the training and test groups are within the same range. The mean percentage of success obtained with the training group for DF (Table 3) was 87% for active (+) and 94% for inactive (−). For the test it was 80% and 92%, respectively. The results were similar to those obtained with the DF equation, which points out the validity of the LDA equation.

The results of the external validation test are shown in Table 4. As can be seen, for the active set there was a misclassified compound, namely ethionamide. The same is found in the opposite group, where glucosamine was misclassified.

Figure 2 shows the PDD obtained from DF. As can be inferred from Figure 2, the optimal range of DF to find active compounds can be established between 0 and 4.5.

### 2.2. Similarity-Based Virtual Screening

A virtual library containing new pyrimidine derivatives generated by combining the chemical scaffolds and substituents depicted in Table 5 was screened as described in Section 3.2. After building the database containing the substituted pyrimidines, the virtual screening was performed. Descriptors calculated for the entire database are available to the readers upon request to the corresponding author. Based on these parameters, a virtual screening was carried out so that those molecular structures with the values of the descriptors and DFs within the thresholds, shown in the Appendix A, were selected as candidates. Appendix A, shows the calculated descriptors for the selected compounds. Seven potentially active molecules were selected. These structures are presented in Figure 3. These compounds selected as possible candidates were synthesized and tested in vitro.

### 2.3. Chemistry

**Synthesis of 1,2,4-triazolo[1,5-*a*]pyrimidine derivatives.** Reaction of the required magnesium acetylides, generated from alkynes **1a-b** and *i*-PrMgCl in dry THF at 0 °C, with piperonal **2**, afforded the expected propargylic alcohols **3a-b**. Oxidation of **3a-b** with MnO_2_ afforded α-acetylene ketones **4a** and **4b** in 79% and 37% overall yield, respectively (Figure 2).

Cyclocondensation of 3-amino-5-benzylsulfanyl-1,2,4-triazole **5** with α-acetylene ketones **4a-b** in dry DMF at 40 °C, gave the corresponding triazolo[1,5-*a*]pyrimidines **6a-b** in moderate yields (73% and 38%, respectively) (Figure 2).

Finally, deprotection of **6b** in acidic conditions afforded **7** in 77% yield (Figure 3).

**Synthesis of pyrimidine derivatives.** Two of us reported on the synthesis of novel 4-alkoxypyrimidines starting from 2-alkylsulfanylpyrimidinones of type **8** [103,104]. The method is based on a selective *O*-alkylation reaction with bulky aliphatic alcohols using the Mitsunobu conditions (method A, Figure 4) or in basic medium with sterically demanding agents like α–haloketones (method B, Figure 4).

Oxidation of the thioether moiety to the corresponding sulfone **10** using *m*-CPBA and nucleophilic displacement by different nucleophiles produced the corresponding highly molecular diverse pyrimidines of type **11**. In addition, when 4-isopropoxipyrimidine **11a** was treated with a 1:1 mixture of H_2_SO_4_/AcOH at 90 °C for 15 min, the selective cleavage of the 4-isopropoxy group took place yielding 2-aryloxypyrimidinone **12** (Figure 5).

On the other hand, the reaction of the deprotected amine **9e** with phenylboronic acid and glyoxylic acid (Petasis reaction) [105,106] gave the desired α-phenyl glycine derivative **13** in moderated yield (Figure 6).

Finally, we prepared the compound **14** by reducing the carbonyl group of compound **9d** with NaBH_4_ in MeOH. This reduction afforded the hydroxy derivative **14** in 80% isolated yield (Figure 7).

**Microbiological study.** To check the predicted antituberculosis activity of the selected candidates, microbiological tests were performed. The result of in vitro susceptibility test of conventional drugs is shown in Table 6, and the experimental results of the selected compounds are illustrated in Table 7. Two compounds, **7** and **12**, showed MIC_50_ = MIC_90_ = 32 mg/L, corresponding to 81.5 μM and 121.1 μM, respectively; four compounds, **6a**, **9c**, **11b** and **13**, showed MIC_90_ = 64 mg/L against *M. tuberculosis*, which corresponds to 146, 167.8, 160.1, 131.8 μM, respectively; and **14** was inactive at the assayed concentrations. The compound **11b** showed the best MIC_50_ = 80 μM, but its MIC_90_ raised to 160,1 μM. Thus, the results pointed out that **7** was the most active agent, showing molar MIC_90_ four times lower than Ethambutol but within the same magnitude order.

## 3. Materials and Methods

### 3.1. Antituberculosis Activity Modelling

A group of 32 compounds with known activity against MTBC were compiled from various sources [96,97,98]. Their structures are shown in Appendix A. A set of 45 compounds with a different pharmacological activity was also used as inactive group. Their structures are shown in Appendix A. The compounds were characterized by a set of 90 descriptors calculated using the DesMol program [99]. The descriptors used with their symbols, definitions and references are shown in Table 1. These were used to build a model capable of discriminating between active and inactive antituberculous compounds.

DFs were calculated with randomly selected subsets of 25 out of 32 active and 35 out of 45 inactive compounds by using BMDP New System [107]. Descriptor selection was based on the Fisher-Snedecor F parameter. The variables were introduced step by step in the DF: in each step, the variable that added the most to the separation of the groups was entered in the equation, or the variable that contributed the least to improving said separation was eliminated. The classification criterion was the shortest Mahalanobis distance. The discriminant ability of the DF was evaluated by two parameters, Wilks λ, and the percentage of correct classification in each group. The independent variables in this study were the calculated structural invariants, and the discrimination property was the presence of activity against MTBC.

The validation of the selected DF was performed by two methods. One internal leave-*n*-out test in which the program randomly chose and pulled out a ratio of compounds, and used them to evaluate the DF obtained with the rest; and another external test with a previously unused data set.

To choose the optimal ranges of values for this equation, the corresponding Pharmacological Distribution Diagram, PDD [108], was obtained. These diagrams are useful to determine the intervals of the equation in which the probability of finding new candidates is maximum. This is a graph similar to a histogram in which expectancies appear on the ordinate axis. For an arbitrary range of values of a given function, the activity expectancy is E_a_ = a/(i + 100), where a is the percentage of active compounds in the range and i is the corresponding percentage of inactive within the same range. The expectancy of inactivity is similarly defined as E_i_ = i/(a + 100). This plot provides a good visualization of the regions of minimal overlap between the active and inactive compounds and helps to select the optimal interval of the DF.

The 90 descriptors and the selected discriminant function were used as filters to select potential candidates in structural databases. For this, the maximum and minimum values of each descriptor were established as thresholds for the 90 variables, while for DF, the optimal interval was taken. Compounds exhibiting all 90 values within the thresholds and DF values within the optimal intervals were considered as candidates.

### 3.2. Similarity-Based Virtual Screening

A virtual library containing 320 structures of substituted pyrimidines was generated by combining the chemical scaffolds and substituents depicted in Table 5, using in-house software. The set of 90 descriptors and DFs were calculated for this library. A virtual screening was carried out based on these parameters to select as candidates those molecular structures whose values for the descriptors and DFs were within the thresholds.

### 3.3. Chemical Methods

DMF was dried over activated molecular sieves (4 Å). THF was dried over Na/benzophenone prior use. All the other commercially available chemicals were used as purchased without further purification. Reactions involving magnesium acetylides and synthesis of triazolo[1,5-*a*]pyrimidines were run under a dry Ar atmosphere. Melting points (capillary tube) were measured with an electrothermal digital melting point apparatus IA 91,000 and are uncorrected. IR spectra were recorded on a Mattson-Galaxy Satellite FT-IR. Additionally, ^1^H and ^13^C NMR spectra were recorded at 200 and 50 MHz, respectively, on a Brucker DPX200 Advance instrument with TMS as internal standard. MS spectra were recorded on a VG Quattro instrument in the positive ionization FAB mode, using 3-NBA or 1-thioglycerol as the matrix or in a Thermo Quest 2000 series apparatus for the EI (70 eV) mode. Analytical TLC was performed on precoated TLC plates, silica gel 60 F_254_ (Merck). Flash-chromatography (FC) purifications were performed on silica gel 60 (230–400 mesh, Merck).


**Synthesis of propargylic alcohols 3a-b. General procedure.**


To a cooled (0 °C) solution of the corresponding alkyne **1a-b** in dry THF (2 mL/mmol), *i*-PrMgCl (2 M solution in THF) was added dropwise under Ar. The mixture was stirred at that temperature for 4 h. Then, a solution of the piperonal **2** (1.3 equiv) in dry THF (1 mL/mmol) was slowly added dropwise over a period of 15 min. The reaction mixture, under Ar, was stirred from 0 °C to r.t. until total consumption of **1a-b** (5–12 h., monitored by TLC). The reaction was quenched with saturated solution of NH_4_Cl (3 mL/mmol) at r.t. and the organic solvent was eliminated under reduced pressure. The aqueous layer was extracted with AcOEt (3 × 3 mL/mmol) and the combined organic layers were dried over MgSO_4_, the solvent was evaporated and the resulting residue purified by flash-chromatography (n-hexane:AcOEt).

**1-Benzo[1,3]dioxol-5-yl-3-phenyl-prop-2-yn-1-ol (3a).** According to the general procedure described above, the reaction between **1a** (4.01 g, 39.26 mmol) and piperonal **2** (7.65 g, 50.9 mmol) afforded 8.02 g (81%) of **3a** as a white solid. m.p.: 60–61 °C. IR (KBr): ν 3439 (br., OH). ^1^H NMR (CDCl_3_): δ 7.5–7.4 (m, 2*H*_arom_), 7.3 (m, 3*H*_arom_), 7.2–7.1 (m, 2*H*_arom_), 6.85 (d, 1*H*_arom_, *J* = 8.0 Hz), 6.00 (s, 2H, OC*H*_2_O), 5.63 (s, 1H, C*H*C≡C), 2.59 (br., 1H, O*H*). ^13^C NMR (CDCl_3_): δ 147.8, 147.6, 134.6 (3s, 3*C*_arom_), 131.7, 128.6, 128.3 (3d, 5*C*H_arom_), 122.2 (s, *C*_arom_), 120.4, 108.1, 107.4 (3d, 3*C*H_arom_), 101.2 (t, *C*H_2_), 88.6 (s, *C*≡CPh), 86.5 (s, C≡*C*Ph), 64.8 (d, *C*H). MS (FAB^+^) *m*/*z*: 253 ([M+1]^+^, 9). Anal. Calcd. for C_16_H_12_O_3_: C, 76.18; H, 4.79. Found: C, 76.39; H, 4.61%.

**1-Benzo[1,3]dioxol-5-yl-3-(tetrahydro-pyran-2-yloxy)-prop-2-yn-1-ol (3b).** According to the general procedure described above, the reaction between **1b** (0.21 g, 1.48 mmol) and piperonal **2** (0.29 g, 1.92 mmol) afforded the crude of the compound **3b**. The reaction mixture was used in the next step of synthesis without further purification.


**Synthesis of α-acetylenic ketones 4a-b. General procedure.**


Over a cooled (0 °C), mechanically stirred suspension of MnO_2_ (5 equiv) in CH_2_Cl_2_ (3 mL/mmol), a solution of the propargyl alcohol **3a** or of the reaction mixture **3b** in CH_2_Cl_2_ (3 mL/mmol) was added dropwise. The reaction mixture was stirred from 0 °C to r.t. until total consumption of **3a-b** (3–12 h., monitored by TLC) and then filtered through a Celite^®^ pad. The solvents were removed under reduced pressure and the resulting residue was purified by flash-chromatography (n-hexane:AcOEt).

**1-Benzo[1,3]dioxol-5-yl-3-phenyl-propynone (4a).** According to the general procedure described above, the reaction between **3a** (1.98 g, 7.87 mmol) and MnO_2_ (3.84 g, 39.35 mmol) afforded after crystallization of the residue with MeOH:H_2_O instead of the chromatographic purification 1.91 g (97%) of **4a** as a yellow solid. m.p.: 100–101 °C. IR (KBr): ν 1625 (s, C=O). ^1^H NMR (CDCl_3_): δ 7.92 (dd, 1*H*_arom_, *J* = 8.2 Hz, *J*′ = 1.8 Hz), 7.7–7.6 (m, 3*H*_arom_), 7.5–7.4 (m, 3*H*_arom_), 6.93 (d, 1*H*_arom_, *J* = 8.2 Hz), 6.10 (s, 2H, OC*H*_2_O). ^13^C NMR (CDCl_3_): δ 176.0 (s, *C*=O), 152.8, 148.1 (2s, 2*C*_arom_), 132.9 (d, 2*C*H_arom_), 131.9 (s, *C*_arom_), 130.6, 128.6, 127.2 (3d, 4*C*H_arom_), 120 (s, *C*_arom_), 108.2, 107.9 (2d, 2*C*H_arom_), 102.1 (t, *C*H_2_), 92.3 (s, C≡*C*Ph), 86.7 (s, *C*≡CPh). MS (FAB^+^) *m*/*z*: 251 ([M+1]^+^, 100). Anal. Calcd. for C_16_H_10_O_3_: C, 76.79; H, 4.03. Found: C, 76.53; H, 4.24%. IR and NMR spectra can be seen in Appendix A.

**1-Benzo[1,3]dioxol-5-yl-4-(tetrahydro-pyran-2-yloxy)-but-2-yn-1-one (4b).** According to the general procedure described above, the reaction between the reaction mixture of **3b** and MnO_2_ (0.72 g, 7.4 mmol) afforded 157 mg (overall yield = 37%) of **4b** as a yellow oil. IR (NaCl): ν 1640 (m, C=O). ^1^H NMR (CDCl_3_): δ 7.82 (dd, 1*H*_arom_, *J* = 7.4 Hz, *J*′ = 1.8 Hz), 7.55 (d, 1*H*_arom_, *J* = 1.6 Hz), 6.89 (d, 1*H*_arom_, *J* = 8.2 Hz), 6.08 (s, 2H, OC*H*_2_O), 4.9–4.8 (m, 1H, C*H*), 4.56 (s, 2H, C*H*_2_OThp), 3.9–3.8 (m, 1H, C*H*_2_), 3.6–3.5 (m, 1H, C*H*_2_), 1.9–1.6 (m, 6H, 3C*H*_2_). ^13^C NMR (CDCl_3_): δ 175.6 (s, *C*=O), 152.9, 148.1 (2s, 2*C*_arom_), 131.5 (s, *C*_arom_), 127.4 (d, *C*H_arom_), 108.2, 107.9 (2d, 2*C*H_arom_), 102.1 (t, *C*H_2_), 97.3 (d, *C*H), 89.6 (s, C≡*C*CH_2_), 83.3 (s, *C*≡CCH_2_), 62.0, 54.0, 30.1, 25.2, 18.8 (5t, 5*C*H_2_). MS (FAB^+^) *m*/*z*: 289 ([M+1]^+^, 100). Anal. Calcd. for C_16_H_16_O_5_: C, 66.66; H, 5.59. Found: 66.94; H, 5.76%. IR and NMR spectra can be seen in Appendix A.


**Synthesis of triazolo[1,5-*a*]pyrimidines 6a-b. General procedure.**


A mixture of 3-amino-5-benzylsulfanyl-1,2,4-triazole **5**, DBU and anhydrous MgSO_4_ (5 g/g) in dry DMF (3 mL/mmol) was heated at 40 °C under Ar for 30 min. Then, a solution of the corresponding α-acetylenic ketone **4a-b** in dry DMF (3 mL/mmol) was slowly added using a *syringe pump* over a period of 5h. The reaction mixture, under Ar, was stirred at 40 °C until total consumption of **5** (1–2 h., monitored by TLC). The reaction was filtered and the organic solvent was eliminated under reduced pressure. The residue was dissolved with CH_2_Cl_2_ (15 mL/mmol) and was washed with saturated solution of NH_4_Cl (3 × 3 mL/mmol). The organic layer was dried over MgSO_4_, the solvent was evaporated and the resulting residue was purified by flash-chromatography (n-hexane:AcOEt).

**5-Benzo[1,3]dioxol-5-yl-2-benzylsulfanyl-7-phenyl-1,2,4-triazolo[1,5-*a*]pyrimidine (6a).** According to the general procedure described above, reaction between **5** (106 mg, 0.52 mmol), **4a** (303 mg, 1.21 mmol, 2.5 equiv), DBU (81 μL, 0.53 mmol, 1.1 equiv) and anhydrous MgSO_4_ (0.5g) afforded 166 mg (73%) of **6a** as a white solid. m.p.: 143–144 °C. ^1^H NMR (DMSO-*d*_6_): δ 8.3–7.1 (m, 14*H*_arom_), 6.21 (s, 2H, OC*H*_2_O), 4.59 (s, 2H, PhC*H*_2_S). ^13^C NMR (DMSO-*d*_6_): δ 166.4, 159.4, 158.1, 150.1, 148.1, 146.1, 137.7 (7s, 8*C*_arom_), 131.5 (d, *C*H_arom_), 130.1 (s, *C*_arom_), 129.7, 128.9, 128.4, 128.3, 127.2, 123.0, 108.5, 107.3, 105.6 (9d, 13*C*H_arom_), 101.8, 34.4 (2t, 2*C*H_2_). MS (FAB^+^) *m*/*z*: 439 ([M+1]^+^, 100). Anal. Calcd. for C_25_H_18_N_4_O_2_S: C, 68.48; H, 4.14; N, 12.78; S, 7.31. Found: C, 68.29; H, 4.27; N, 13.01; S, 7.02%. IR and NMR spectra can be seen in Appendix A.

**5-Benzo[1,3]dioxol-5-yl-2-benzylsulfanyl-7-(tetrahydro-pyran-2-yloxymethyl)-1,2,4-triazolo[1,5-*a*]pyrimidine (6b).** According to the general procedure described above, the reaction between **5** (255 mg, 1.24 mmol), **4b** (698 mg, 2.42 mmol, 2 equiv), DBU (12 μL, 0.082 mmol, 5% mol.) and anhydrous MgSO_4_ (1.25 g,) afforded 225 mg (38%) of **6b** as a yellow solid. m.p.: 62–64 °C. ^1^H NMR (CDCl_3_): δ 7.8–6.9 (m, 9*H*_arom_), 6.09 (s, 2H, OC*H*_2_O), 5.24 (d, 1H, *J* = 17 Hz, C*H*_2_OThp), 5.01 (d, 1H, *J* = 17 Hz, C*H*_2_OThp), 4.9 (m, 1H, C*H*), 4.60 (s, 2H, PhC*H*_2_S), 4.0–3.9 (m, 1H, C*H*_2_), 3.7–3.6 (m, 1H, C*H*_2_), 1.9–1.7 (m, 6H, 3C*H*_2_). ^13^C NMR (CDCl_3_): δ 168.3, 160.1, 155.8, 150.4, 148.5, 146.5, 137.2, 130.7 (8s, 8*C*_arom_), 129.1, 128.5, 127.4, 122.7, 108.4, 107.8, 102.7 (7d, 9*C*H_arom_), 101.7 (t, *C*H_2_), 99.14 (d, *C*H), 62.6, 62.5, 35.6, 30.2, 25.1, 19.3 (6t, 6*C*H_2_). MS (FAB^+^) *m*/*z*: 477 ([M+1]^+^, 100). Anal. Calcd. for C_25_H_24_N_4_O_4_S: C, 63.01; H, 5.08; N, 11.76; S, 6.73. Found: C, 63.22; H, 4.84; N, 11.92; S, 6.50%.

**Synthesis of (5-benzo[1,3]dioxol-5-yl-2-benzylsulfanyl-1,2,4-triazolo[1,5-*a*]pyrimidin-7-yl)-methanol (7).** A solution of 5-benzo[1,3]dioxol-5-yl-2-benzylsulfanyl-7-(tetrahydro-pyran-2-yloxymethyl)-1,2,4-triazolo[1,5-*a*]pyrimidine **6b** (99 mg, 0.21 mmol) in (AcOH:THF:H_2_O) (4:2:1) (4 mL/mmol, 1 mL) was heated at 45 °C for 19 h. The organic solvent was eliminated under reduced pressure, was added saturated solution of NaHCO_3_ (3 mL/mmol) and the aqueous layer was extracted with AcOEt (3×3 mL/mmol). The combined organic layers were dried over MgSO_4_, the solvent was evaporated and the resulting residue purified by flash-chromatography (n-hexane:AcOEt) afforded 64 mg (77%) of **7** as a yellow solid. m.p.: 175–177 °C. IR (KBr): ν 3199 (br., OH). ^1^H NMR (DMSO-*d*_6_): δ 7.9–7.2 (m, 9*H*_arom_), 6.25 (s, 2H, OC*H*_2_O), 6.17 (t, 1H, *J* = 5.5 Hz, CH_2_O*H*), 5.05 (d, 2H, *J* = 5.5 Hz, C*H*_2_OH), 4.63 (s, 2H, PhC*H*_2_S). ^13^C NMR (DMSO-*d*_6_): δ 166.6, 159.3, 155.2, 150.5, 150.1, 148.3, 137.7, 130.2 (8s, 8*C*_arom_), 128.9, 128,4, 127.2, 122.7, 108.7, 107.0, 102.5 (7d, 9*C*H_arom_), 101.9, 57.7, 34.4 (3t, 3*C*H_2_). MS (FAB^+^) *m*/*z*: 393 ([M+1]^+^, 100). Anal. Calcd. for C_20_H_16_N_4_O_3_S: C, 61.21; H, 4.11; N, 14.28; S, 8.17. Found: C, 60.94; H, 4.36; N, 14.09; S, 8.39%.


**Method A for the preparation of the alkoxypyrimidines 9a and 9e. Mitsunobu reaction.**


A solution of DIAD (1.2 equiv) in dry THF (1 mL/mmol) is added dropwise to a solution of Ph_3_P (1.2 equiv), the appropriate 2-benzylsulfanylpyrimidinone **8a-b** (1 equiv) and different alcohols in tetrahydrofuran (2 mL/mmol) at room temperature. The reaction mixture was stirred at room temperature until total disappearance of **8a-b** (TLC monitoring). The solvent was evaporated until dryness and the crude product adsorbed over silica purified by flash chromatography (n-hexane:EtOAc).

**2-Benzylsulfanyl-4-isopropoxy-6-phenyl-pyrimidine (9a).** According to the general procedure described above, the reaction between **8b** (500 mg, 1.70 mmol), TPP (675 mg, 2.55 mmol), and DIAD (0.50 mL, 2.55 mmol) in dry THF (6 mL), afforded 526 mg (92%) of **9a** isolated as colorless solid after 2 h. m.p.: 81–82 °C. ^1^H NMR (CDCl_3_): δ 8.1–8.0 (s, 2*H*_arom_), 7.7–7.2 (m, 8*H*_arom_), 6.77 (s, 1H, *H*_pyrim_), 5.46 (hept, 1H, *J* = 6.2 Hz, C*H*(CH_3_)_2_), 4.54 (s, 2H, PhC*H*_2_S), 1.38 (d, 6H, *J* = 6.2 Hz, CH(C*H*_3_)_2_). ^13^C NMR (CDCl_3_): δ 170.8, 169.4, 164.6 (3s, 3*C*_pyrim_), 138.0, 136.8 (2s, 2*C*_arom_), 130.5, 128.8, 128.7, 128.5, 128.4, 127.0 (6d, 10*C*H_arom_), 99.7 (d, *C*H_pyrim_), 69.5 (d, *C*H), 35.4 (t, *C*H_2_), 21.9 (q, 2*C*H_3_). MS (EI) *m*/*z*: 336 ([M]^·+^, 90). Anal. Calcd. for C_20_H_20_N_2_OS: C, 71.40; H, 5.99; N, 8.33; S, 9.53. Found: C, 71.51; H, 6.17; N, 8.14; S, 9.25%.

**[1-Benzyl-2-(2-benzylsulfanyl-pyrimidin-4-yloxy)-ethyl]-carbamic acid *tert*-butyl ester (9e).** According to the general procedure described above, the reaction between **8a** (1.00 g, 4.59 mmol), TPP (1.58 g, 5.96 mmol), *N*-Boc-phenylalaninol (1.50 g, 5.96 mmol) and DIAD (1.15 mL, 5.96 mmol) in dry THF (15 mL), afforded 1.47 g (71%) of **9e** isolated as colorless solid after 5 h. m.p.: 108–110 °C. IR (KBr): ν 3388 (br., NH), 1685 (s, C=O). ^1^H NMR (CDCl_3_): δ 8.30 (d, 1H, *J* = 4.8 Hz, *H*_pyrim_), 7.5–7.4 (m, 10*H*_arom_), 6.48 (d, 1H, *J* = 4.8 Hz, *H*_pyrim_), 4.81 (br., 1H, N*H*), 4.37 (s, 2H, C*H*_2_), 4.30 (s, 2H, C*H*_2_), 4.25 (br., 1H, C*H*), 2.90 (m, 2H, PhC*H*_2_C), 1.39 (s, 9H, (C*H*_3_)_3_). ^13^C NMR (CDCl_3_): δ 171.3, 168.4 (2s, 2*C*_pyrim_), 157.5 (d, CH_pyrim_), 152.2 (s, *C*=O), 137.8, 137.3 (2s, 2*C*_arom_), 129.3, 128.8, 128.5, 127.1, 126.6, 126.4 (6d, 10*C*H_arom_), 103.7 (d, *C*H_pyrim_), 79.6 (s, *C*), 64.1 (t, *C*H_2_), 50.8 (d, *C*H), 37.8, 35.2 (2t, 2*C*H_2_), 28.3 (q, 3*C*H_3_). MS (FAB^+^) *m*/*z*: 452 ([M+1]^+^, 17). Anal. Calcd. for C_25_H_29_N_3_O_3_S: C, 66.49; H, 6.47; N, 9.31; S, 7.10. Found: C, 66.30; H, 6.68; N, 9.07; S, 6.08%.


**Method B for the preparation of the alkoxypyrimidines 9b-d.**


To a solution of the corresponding 2-benzylsulfanylpyrimidinone **8a-b** (1 equiv) in DMF (3 mL per mmol), 1.1 equiv of TMG were added. Then, the corresponding phenacyl bromide (1.1. equiv) was added dropwise. The reaction mixture was until total disappearance of **8a-b** (TLC monitoring). The solvent was evaporated until dryness and the crude product adsorbed over silica purified by flash chromatography (n-hexane:EtOAc).

**2-(2-Benzylsulfanyl-pyrimidin-4-yloxy)-1-(4-chloro-phenyl)-ethanone (9b).** According to the general procedure described above, the reaction between **8a** (600 mg, 2.8 mmol), (4-chloro-phenyl)-acetyl bromide (850 mg, 3.6 mmol) and TMG (0.45 mL, 3.6 mmol) in dry DMF (8 mL), afforded 856 mg (83%) of **9b** isolated as colorless solid after 6 h. m.p.: 111–112 °C. IR (KBr): ν 1698 (s, C=O). ^1^H NMR (CDCl_3_): δ 8.34 (d, 1H, *J* = 5.6 Hz, *H*_pyrim_), 7.9–7.3 (m, 9*H*_arom_), 6.64 (d, 1H, *J* = 5.6 Hz, *H*_pyrim_), 5.54 (s, 2H, C*H*_2_O), 4.25 (s, 2H, PhC*H*_2_S). ^13^C NMR (CDCl_3_): δ 191.7 (s, *C*=O), 171.0, 167.6 (2s, 2*C*_pyrim_), 157.8 (d, *C*H_pyrim_), 140.3, 137.0, 132.5 (3s, 3*C*_arom_), 129.2, 128.5, 128.4, 127.1, (4d, 9*C*H_arom_), 103.8 (d, *C*H_pyrim_), 67.4, 35.1 (2t, 2*C*H_2_). MS (EI) *m*/*z*: 372 ([M+2]^·+^, 19), 370 ([M]^·+^, 52). Anal. Calcd. for C_19_H_15_ClN_2_O_2_S: C, 61.53; H, 4.08; N, 7.55; S, 8.65. Found: C, 61.32; H, 3.98; N, 7.76; S, 8.46%.

**2-(2-Benzylsulfanyl-pyrimidin-4-yloxy)-1-(3-nitro-phenyl)-ethanone (9c).** According to the general procedure described above, the reaction between **8a** (1.00 g, 4.6 mmol), (3-nitro-phenyl)-acetyl bromide (1.64 g, 5.7 mmol) and TMG (0.72 mL, 5.7 mmol) in dry DMF (15 mL), afforded 1.31 g (75%) of **9c** isolated as colorless solid after 6 h. m.p.: 97–98 °C. IR (KBr): ν 1705 (s, C=O). ^1^H NMR (CDCl_3_): δ 8.74 (s, 1*H*_arom_), 8.46 (d, 1*H*_arom_, *J* = 8.2 Hz), 8.35 (d, 1H, *J* = 5.6 Hz, *H*_pyrim_), 8.24 (d, 1*H*_arom_, *J* = 7.8 Hz), 7.71 (t, 1*H*_arom_), 7.3–7.2 (m, 5*H*_arom_), 6.65 (d, 1H, *J* = 5.6 Hz, *H*_pyrim_), 5.60 (s, 2H, C*H*_2_O), 4.26 (s, 2H, PhC*H*_2_S). ^13^C NMR (CDCl_3_): δ 191.1 (s, *C*=O), 171.1, 167.4 (2s, 2*C*_pyrim_), 158.0 (d, *C*H_pyrim_), 148.4, 137.0, 135.4 (3s, 3*C*_arom_), 133.3, 130.2, 128.5, 128.3, 128.0, 127.1, 122.7 (7d, 9*C*H_arom_), 103.8 (d, *C*H_pyrim_), 67.5, 35.0 (2t, 2*C*H_2_). MS (EI) *m*/*z*: 381 ([M]^·+^, 18). Anal. Calcd. for C_19_H_15_N_3_O_4_S: C, 59.83; H, 3.96; N, 11.02; S, 8.41. Found: C, 59.57; H, 4.11; N, 11.16; S, 8.14%.

**2-(2-Benzylsulfanyl-6-phenyl-pyrimidin-4-yloxy)-1-phenyl-ethanone (9d).** According to the general procedure described above, the reaction between **8b** (500 mg, 1.7 mmol), phenyl-acetyl bromide (415 mg, 2.0 mmol) and TMG (0.26 mL, 2.0 mmol) in dry DMF (5 mL), afforded 512 mg (73%) of **9d** isolated as colorless solid after 4 h. m.p.: 134–135 °C. IR (KBr): 1706 (m, C=O). ^1^H NMR (CDCl_3_): δ 8.1–8.0 (m, 4*H*_arom_), 7.6–7.5 (m, 6*H*_arom_), 7.4–7.1 (m, 5*H*_arom_), 7.05 (s, 1H, *H*_pyrim_), 5.67 (s, 2H, OC*H*_2_), 4.39 (s, 2H, PhC*H*_2_S). ^13^C NMR (CDCl_3_): δ 192.9 (s, *C*=O), 170.7, 168.9, 165.2 (3s, 3*C*_pirim_), 137.5, 136.4, 134.3 (3s, 3*C*_arom_), 133.8, 130.7, 128.8, 128.7, 128.6, 128.4, 127.7, 127.1, 127.0 (9d, 15*C*H_arom_), 99.1 (d, *C*H_pyrim_), 67.7, 35.2 (2t, 2*C*H_2_). MS (EI) *m*/*z*: 412 ([M]^+^, 25). Anal. Calcd. for C_25_H_20_N_2_O_2_S: C, 72.79; H, 4.89; N, 6.79; S, 7.77. Found: C, 72.57; H, 5.00; N, 6.56; S, 8.09%.


**Oxidation of 9a,b to Sulfones 10a,b. General Procedure.**


To a cooled (0 °C) solution of pyrimidine derivatives **9a,b** (1 equiv) in CH_2_Cl_2_ (5 mL per mmol), 2.5 equiv of *m*-CPBA (60% purity) were added in small portions. The mixture was then stirred at 0 °C during 2 h, then diluted with CH_2_Cl_2_ (20 mL per mmol) and washed with aq. satd. NaHCO_3_ solution (2×5 mL per mmol) and brine (5 mL per mmol). The separated organic layer was dried (MgSO_4_), filtered and evaporated to give a residue which was purified by flash chromatography using n-hexane/EtOAc.

**4-Isopropoxy-6-phenyl-2-phenylmethanesulfonyl-pyrimidine (10a).** According to the general procedure described above, the reaction of **9a** (450 mg, 1.34 mmol) and *m*-CPBA (1.15 g, 3.34 mmol) in CH_2_Cl_2_ (7 mL), afforded to **10a** (432 mg, 88%) isolated as a colorless solid. m.p.: 124–125 °C. ^1^H NMR (CDCl_3_): δ 8.1–8.0 (m, 2*H*_arom_), 7.6–7.2 (m, 8*H*_arom_), 6.99 (s, 1H, *H*_pyrim_), 5.65 (hept, 1H, *J* = 6.0 Hz, C*H*(CH_3_)_2_), 4.88 (s, 2H, PhC*H*_2_S), 1.43 (d, 6H, *J* = 6.2 Hz, CH(C*H*_3_)_2_). ^13^C NMR (CDCl_3_): δ 170.9, 165.5, 164.7 (3s, 3*C*_pyrim_), 135.0 (s, *C*_arom_), 131.6, 131.3, 128.8, 128.7, 128.2, 127.2 (6d, 10*C*H_arom_), 127.1 (s, *C*_arom_), 106.0 (d, *C*H_pyrim_), 71.6 (d, *C*H), 57.3 (t, *C*H_2_), 21.7 (q, 2*C*H_3_). MS (EI) *m*/*z*: 368 ([M]^·+^, 67). Anal. Calcd. for C_20_H_20_N_2_O_3_S: C, 65.20; H, 5.47; N, 7.60; S, 8.70. Found: C, 65.29; H, 5.60; N, 7.48; S, 8.88%.

**1-(4-Chloro-phenyl)-2-(2-phenylmethanesulfonyl-pyrimidin-4-yloxy)-ethanone (10b).** According to the general procedure described above, the reaction of **9b** (675 mg, 1.82 mmol) and *m*-CPBA (1.31 g, 4.55 mmol) in CH_2_Cl_2_ (9 mL), afforded to **10b** (613 mg, 84%) isolated as a colorless solid. m.p.: 141–142 °C. IR (KBr): ν 1692 (s, C=O). ^1^H NMR (CDCl_3_): δ 8.65 (d, 1H, *J* = 5.8 Hz, *H*_pyrim_), 7.92 (d, 2*H*_arom_, *J* = 6.8 Hz), 7.52 (d, 2*H*_arom_, *J* = 6.8 Hz), 7.3–7.2 (m, 5*H*_arom_), 7.12 (d, 1H, *J* = 5.6 Hz, *H*_pyrim_), 5.72 (s, 2H, C*H*_2_O), 4.54 (s, 2H, PhC*H*_2_S). ^13^C NMR (CDCl_3_): δ 190.9 (s, *C*=O), 169.3, 163.9 (2s, 2*C*_pyrim_), 158.3 (d, *C*H_pyrim_), 140.8, 132.2 (2s, 2*C*_arom_), 131.0, 129.4, 129.2, 128.8, 128.7 (5d, 9*C*H_arom_), 126.4 (s, *C*_arom_), 111.4 (d, *C*H_pyrim_), 68.3, 57.6 (2t, 2*C*H_2_). MS (FAB^+^) *m*/*z*: 405 ([M+3]^+^, 24), 403 ([M+1]^+^, 62). Anal. Calcd. for C_19_H_15_ClN_2_O_4_S: C, 56.65; H, 3.75; N, 6.95; S, 7.96. Found: C, 56.81; H, 3.54; N, 7.18; S, 8.15%.


***Ipso*-substitution reaction of pyrimidinyl sulfone derivatives 10a,b.**


Synthesis of 4-isopropoxy-2-phenoxy-6-phenyl-pyrimidine (11a). To a solution of phenol (55 mg, 0.57 mmol) in dioxane (2 mL), the Cs_2_CO_3_ (210 mg, 0.59 mmol) was added. The reaction mixture was stirred at r.t. for 15–20 min. Then, the sulfone 10a (200 mg, 0.54 mmol) was added. After stirring at 60 °C for 3 hours, the solvent was removed *in vacuo* and the mixture was acidified with 2 *N* hydrochloric acid and extracted with ethyl acetate. The residue was purified using flash chromatography to give 11a (131 mg, 80%) as a colorless solid. m.p.: 68–69 °C. ^1^H NMR (CDCl_3_): δ 8.0–7.9 (m, 2*H*_arom_), 7.5–7.3 (m, 8*H*_arom_), 6.84 (s, 1H, *H*_pyrim_), 5.28 (hept, 1H, *J* = 6.2 Hz, C*H*(CH_3_)_2_), 1.34 (d, 6H, *J* = 6.2 Hz, CH(C*H*_3_)_2_). ^13^C NMR (CDCl_3_): δ 171.9, 166.4, 165.0 (3s, 3*C*_pyrim_), 153.2, 136.5 (2s, 2*C*_arom_), 130.6, 129.1, 128.7, 127.0, 124.8, 121.9 (6d, 10*C*H_arom_), 98.6 (d, *C*H_pyrim_), 69.9 (d, *C*H), 21.8 (q, 2*C*H_3_). MS (EI) *m*/*z*: 306 ([M]^·+^, 24). Anal. Calcd. for C_19_H_18_N_2_O_2_: C, 74.49; H, 5.92; N, 9.14. Found: C, 74.36; H, 5.75; N, 9.40%.

**Synthesis of 1-(4-chloro-phenyl)-2-[2-(2,4-dimethoxy-phenylamino)-pyrimidin-4-yloxy]-ethanone (11b).** To a solution of the sulfone **10b** (125 mg, 0.31 mmol) in dioxane (2 mL), the 2,4-dimethoxyaniline (100 mg, 0.62 mmol) was added. The reaction mixture with good stirring was heated at 100 °C until total consumption of the starting material (30 h., TLC monitoring). The solvent was removed under reduced pressure and the residue purified by flash-chromatography (n-hexane:EtOAc) to give **11b** (25 mg, 20%) as an orange crystalline solid. m.p.: 122–123 °C. IR (KBr): ν 3244 (br., NH), 1705 (m, C=O). ^1^H NMR (CDCl_3_): δ 8.20 (d, 1H, *J* = 5.4 Hz, *H*_pyrim_), 7.96 (d, 2*H*_arom_, *J* = 8.2 Hz), 7.77 (d, 1*H*_arom_, *J* = 8.8 Hz), 7.54 (d, 2*H*_arom_, *J* = 8.2 Hz), 7.30 (s, 1*H*_arom_), 6.44 (d, 1*H*_arom_, *J* = 2.4 Hz), 6.36 (d, 1H, *J* = 5.2 Hz, *H*_pyrim_), 5.90 (d, 1H, *J* = 8.2 Hz, N*H*), 5.58 (s, 2H, C*H*_2_O), 3.83 (s, 3H, C*H*_3_O), 3.72 (s, 3H, C*H*_3_O). ^13^C NMR (CDCl_3_): δ 192.2 (s, *C*=O), 168.5, 159.6 (2s, 2*C*_pyrim_), 158.6 (d, *C*H_pyrim_), 155.4, 149.7, 140.2, 132.9 (4s, 4*C*_arom_), 129.3, 129.2 (2d, 4*C*H_arom_), 121.9 (s, *C*_arom._), 120.3, 102.7 (2d, 2*C*H_arom_), 98.6 (d, *C*H_pyrim_), 98.5 (d, *C*H_arom_), 67.3 (t, *C*H_2_), 55.6, 55.4 (2q, 2*C*H_3_). MS (EI) *m*/*z*: 401 ([M+2]^·+^, 17), 399 ([M]^·+^, 52). Anal. Calcd. for C_20_H_18_ClN_3_O_4_: C, 60.08; H, 4.54; N, 10.51. Found: C, 60.29; H, 4.48; N, 10.32%.

**Removal of the 4-isopropoxy group. Synthesis of 2-phenoxy-6-phenyl-3*H*-pyrimidin-4-one (12).** The 4-isopropoxypyrimidine **11a** (87 mg, 0.28 mmol) was added to a mixture of AcOH (0.6 mL) and con. H_2_SO_4_ (0.6 mL). The reaction mixture was stirred at 90 °C for 15 min. After cooling, the mixture was neutralized with aq. 5 *N* NaOH and extracted with CH_2_Cl_2_ (3 × 5 mL). The combined organic layers were washed with brine and the separated organic layer was dried (MgSO_4_), filtered and eliminated under reduced pressure to afford the pure pyrimidinone **12** (59 mg, 80%) as a colorless solid. m.p.: 263–264 °C. IR (KBr): ν 3060–2750 (br., NH), 1669 (s, C=O). ^1^H NMR (DMSO-*d*_6_): δ 12.75 (br., N*H*), 7.9–7.8 (m, 2*H*_arom_), 7.6–7.5 (m, 8*H*_arom_), 6.74 (s, 1H, *H*_pyrim_). ^13^C NMR (DMSO-*d*_6_): δ 166.0, 161.4, 158.7 (3s, 3*C*_pyrim_), 151.8, 135.8 (2s, 2*C*_arom_), 130.6, 129.5, 128.8, 126.6, 125.7, 121.7 (6d, 10*C*H_arom_), 102.4 (d, *C*H_pyrim_). MS (EI) *m*/*z*: 264 ([M]^·+^, 54). Anal. Calcd. for C_16_H_12_N_2_O_2_: C, 72.72; H, 4.58; N, 10.70. Found: C, 72.47; H, 4.81; N, 10.86%.


**Synthesis of [1-benzyl-2-(2-benzylsulfanyl-pyrimidin-4-yloxy)-ethylamino]-phenyl-acetic acid (13): Petasis reaction.**


To a stirred solution of glyoxylic acid monohydrate (74 mg, 0.78 mmol) in dichloromethane (5 mL) was added the primary amine **9e**, previous deprotection of Boc using standard conditions, (275 mg, 0.78 mmol), followed by phenylboronic acid (99 mg, 0.78 mmol). After the flask was purged with nitrogen and sealed, the reaction mixture was stirred vigorously at room temperature for 3 days. The resulting precipitate was isolated by filtration and washed with to dichloromethane to give the pure pyrimidinone **13** (197 mg, 52%) as a colorless solid. m.p.: 134–135 °C. IR (KBr): ν 3217 (br, NH), 1713 (s, C=O). ^1^H NMR (DMSO-*d*_6_): δ 8.42 (d, 1H, *J* = 5.8 Hz, *H*_pyrim_), 7.9–7.8 (m, 2H, N*H* + COO*H*), 7.5–7.2 (m, 15*H*_arom_), 6.69 (d, 1H, *J* = 5.8 Hz, *H*_pyrim_), 4.63 (s, 1H, C*H*COOH), 4.34 (s, 2H, PhC*H*_2_S), 4.31 (dd, 1H, *J* = 4.0 Hz, *J*′ = 11.0 Hz, OC*H*_2_), 4.17 (dd, 1H, *J* = 6.0 Hz, *J*′ = 11.0 Hz, OC*H*_2_), 3.15 (br., 1H, PhCH_2_C*H*), 3.02 (dd, 1H, *J* = 4.8 Hz, *J*′ = 13.6 Hz, PhC*H*_2_CH), 2.85 (dd, 1H, *J* = 7.5 Hz, *J*′ = 13.6 Hz, PhC*H*_2_CH). ^13^C NMR (DMSO-*d*_6_): δ 172.6, 170.0, 168.0 (3s, 3*C*), 158.0 (d, *C*H_pyrim_), 137.8, 137.7 (2s, 3*C*_arom_), 134.1, 130.0, 129.2, 128.8, 128.4, 127.8, 127.3, 127.0, 126.4 (9d, 15*C*H_arom_), 104.1 (d, *C*H_pyrim_), 66.9 (t, *C*H_2_), 62.4, 55.5 (2d, 2*C*H), 36.4, 34.2 (2t, 2*C*H_2_). MS (FAB^+^) *m*/*z*: 486 ([M+1]^·+^, 28). Anal. Calcd. for C_28_H_27_N_3_O_3_S: C, 69.25; H, 5.60; N, 8.65; S, 6.60. Found: C, 69.52; H, 5.84; N, 8.48; S, 6.31%.

**Synthesis of 2-(2-benzylsulfanyl-6-phenyl-pyrimidin-4-yloxy)-1-phenyl-ethanol (14).** To a stirred and cooled (0 °C) solution of pyrimidinone **9d** (500 mg, 1.21 mmols) in MeOH (6 mL) was added NaBH_4_ (165 mg, 4.24 mmols) in small portions while stirring (vigorous evolution of gas observed). Stirring was continued for 2 h at 0 °C. The solution was evaporated to dryness, and the crude residue was partitioned between EtOAc (10 mL) and aq satd NH_4_Cl solution (15 mL). The organic layer was separated, washed with H_2_O (5 mL), dried (MgSO_4_), and evaporated to give a residue which was purified by flash chromatography using hexanes/EtOAc to afford pure (**14**) as a colorless solid (403 mg, 80%). m.p.: 140–141 °C. IR (KBr): 3400 (br., OH). ^1^H NMR (CDCl_3_): δ 8.1–8.0 (m, 2*H*_arom_), 7.5–7.3 (m, 13*H*_arom_), 6.90 (s, 1H, *H*_pyrim_), 5.17 (dd, 1H, *J* = 8.4 Hz, *J*′ = 3.2 Hz, PhC*H*), 4.53 (s, 2H, PhC*H*_2_S), 4.47 (dd, 1H, *J* = 11.6 Hz, *J*′ = 8.4 Hz, C*H*_2_O), 4.03 (dd, 1H, *J* = 11.4 Hz, *J*′ = 3.2 Hz, C*H*_2_O), 2.90 (br., 1H, O*H*). ^13^C NMR (CDCl_3_): δ 170.9, 169.6, 165.1 (3s, 3*C*_pyrim_), 139.8, 137.7, 136.4 (3s, 3*C*_arom_), 130.8, 128.8, 128.7, 128.6, 128.5, 128.4, 128.2, 127.1, 126.2 (9d, 15*C*H_arom_), 99.1 (d, *C*H_pyrim_), 72.5 (d, *C*H), 71.7, 35.4 (2t, 2*C*H_2_). MS (EI) *m*/*z*: 414 ([M]^+^, 33). Anal. Calcd. for C_25_H_22_N_2_O_2_S: C, 72.44; H, 5.35; N, 6.76; S, 7.74. Found: C, 72.62; H, 5.14; N, 6.52; S, 7.46%.

### 3.4. Microbiological Methods

Thirty-two isolates of *M. tuberculosis* from respiratory (n = 21) and non-respiratory (n = 11) clinical samples, identified by conventional methods [109] and by DNA hybridization probe (Accuprobe^®^ Gen Probe Inc., San Diego, CA, USA) were selected from the laboratory collection of Dept. Microbiología, Hospital Clínico Universitario. Valencia, Spain. Susceptibility to first-line antituberculosis drugs (ethambutol, isoniazid, rifampin and streptomycin) was tested by a fluorometric method (Bactec^®^ MGIT 960, Becton-Dickinson, Franklin Lakes, NJ, USA) and by a microdilution method [109].

The organisms were grown in modified Middlebrook 7H9 broth supplemented with 10% OADC enrichment (Difco Laboratories, Franklin Lakes, NJ, USA) for seven days at 37 °C. The inoculum size was obtained by dilution of *M. tuberculosis* isolates suspensions in 7H9 broth to yield an absorbance equivalent to that of a MacFarland nº 0.5 standard.

Antimicrobial susceptibility test was performed in 96-well microplates using serial twofold microdilution in 7H9 broth. Initial drug dilutions were prepared in deionized water or, if not soluble, dimethyl sulfoxide. Subsequent twofold dilutions were performed in 150 μL of modified 7H9 broth in the microplates to provide a final test range of 128 to 0.125 mg/L. Ten μL of a suspension of mycobacteria were added to the wells. Plates were covered with Parafilm “M”^®^ (Laboratory Film, American National Can™, Chicago, IL, USA), and incubated for 12 days at 37 °C. Starting at day 13 of incubation, 20 μL of Resazurina^®^ (Sigma 2127, St. Louis, MO, USA) with a concentration of 250 mg/L were added to the wells, and the microplates were reincubated at 37 °C for an additional period of 48 h. MIC_50_ and MIC_90_ were determined as the lowest concentrations of the compounds yielding no visible changes from blue to pink [110].

## 4. Conclusions

New chemical scaffolds have been identified that could render new lead drugs in this field, by using easy to calculate descriptors, such as structural invariants. The only substructure common to all the selected molecules is the pyrimidine ring, and the most frequent substituent is the benzylsulfanyl in position 2. The 1,2,4-triazolo[1,5-*a*]pyrimidine system is present in structures **7** and **6a**, which have the same substituted groups in 2 and 5. Only one pyrimidone was selected (**12**). All these structures are proposed as new base structures in order to design new combinatorial synthesis projects.

Although several action mechanisms are represented in the training group, the usefulness of the method is appreciable. One possible explanation of this fact could be that the equation retains the structural features involved in all the mechanisms considered. In the opinion of the authors, this approach is not able to find new action mechanisms, but it is possible to obtain new unexpected molecular structures acting through the known mechanisms which combine the properties of the known compounds. This feature could be useful in order to avoid problems of resistance.

## Data Availability

Data available upon request to the authors.

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
