# Peer review of "Similarity-Based Virtual Screening to Find Antituberculosis Agents Based on Novel Scaffolds: Design, Syntheses and Pharmacological Assays"

_ijms, 2022, doi:10.3390/ijms232315057_

Round 1
Reviewer 1 Report
This manuscript contains three parts, the structure-property relationship study, the syntheses of some compounds and the microbiological assay of few selected compounds. However, it is hard to see the correlation between the calculated categories and the microbiological results. The current version is not proper for publication. So, reject!
1. Page 3 Line 50: the DF is not explained when it appears at the first time, until the equation of DF in Line 128.
2. Page 3 Line 90: It is better to provide the molecular structures of the 32 compounds and the 45 compounds in a supplementary file.
4. Page 3 Line 98: "with randomly selected subsets of 25 active and 35 inactive compounds". It is not clear what set was used to select the two subsets.
5. Page 3 Line 126: Does LDA stand for the linear discriminant analysis?
6. Table 1: The correlation coefficients between these descriptors should be calculated to check if they are independent to each other. Is it necessary to provide those descriptors which are not used in the only DF equation? It is better to place them in a supplementary file.
7. Table 2: The calculations of the probability-active and probability-inactive should be explained, such as using a formula.
8. Table 2: The chemical structures of these compounds should be provided somewhere, at least in a supplementary material.
9. Table 5: It is not easy to match the scaffold with the fragments. It is better to mark these structures using name or number. How many structures have been generated in this virtual library? More details need to show the results that lead to the selected compounds,
10. The unit of micro-molar concentration is shown in a wrong font (?M), in Table 6 and Page 14 Lines 238-242.
Author Response
Response to Reviewer 1 Comments
Point 1: This manuscript contains three parts, the structure-property relationship study, the syntheses of some compounds and the microbiological assay of few selected compounds. However, it is hard to see the correlation between the calculated categories and the microbiological results. The current version is not proper for publication. So, reject!
Response 1: This objection refers to the section 2.2. ‘Similarity-based Virtual screening’, and can be easily amended. In this section, new text has been included notifying that a Table S5 is included in the Supplementary Materials with the parameters calculated for the selected compounds, where it can be checked that all the parameters, including DF, are within the thresholds established in Table S4. If the reviewer deems it appropriate, we can add another table with these same parameters calculated for a subset of the scanned database, where it is seen that these molecules do not meet any of the thresholds.
Point 2: Page 3 Line 50: the DF is not explained when it appears at the first time, until the equation of DF in Line 128.
Response 2: A short explanation of LDA is given in p.3, that also introduces the discriminant function concept: ‘Given a population, for example of molecules, that can be classified into several groups according to their experimental properties, for example a group of molecules with a pharmacological activity and another without this activity, Linear Discriminant Analy-sis (LDA) is a method to find linear combinations of independent variables (for example the aforementioned structural invariants) whose numerical values can be used to distin-guish between these different categories. When two categories are defined, the classification is done by the so-called discriminant function (DF).’
Point 3: Page 3 Line 90: It is better to provide the molecular structures of the 32 compounds and the 45 compounds in a supplementary file.
Response 3: The structures are provided in Supplementary Tables S1 and S2, respectively.
Point 4: Page 3 Line 98: "with randomly selected subsets of 25 active and 35 inactive compounds". It is not clear what set was used to select the two subsets.
Response 4: The 25 active compounds were selected out of the group of 32, and the 35 inactive ones out of the group of 45. Now this is clarified in the text. Page 3 Line 105.
Point 5: Page 3 Line 126: Does LDA stand for the linear discriminant analysis?
Response 5: Yes. Now it is explained in the text. Page 3 Line 101.
Point 6: Table 1: The correlation coefficients between these descriptors should be calculated to check if they are independent to each other. Is it necessary to provide those descriptors which are not used in the only DF equation? It is better to place them in a supplementary file.
Response 6: The intercorrelations have been included in the Table S4, where the strongest correlation was 0.424, which guarantees the independence of the variables. This is also commented in the text now. Page 7 Line 158.
We think it is better do not split the table 1, because its full version gives a more complete vision of the method employed, because the definitions of the selected indexes require the definitions of other indexes of the table to be understood, and because there is already too much supplementary material.
Point 7: Table 2: The calculations of the probability-active and probability-inactive should be explained, such as using a formula.
Response 7: These probabilities are the so called posterior probabilities, computed by the Bayes rule as the probability of classifying a case (molecule) conditioned to the model obtained.
Let πk the prior probability:
The posterior probability P is given by:
where is the class-conditional density of the case x in the class k.
Assuming that this density for x, given every class k, follows a normal distribution, the density formula for a multivariate Gaussian distribution is applicable. Thus,
where x and the mean are both column vectors,
Cov is the covariance matrix and p is its dimension.
The denominator involves the square root of the determinant of this matrix.
The result of the matrix multiplications in the numerator is a scalar number.
This explanation has been included in a note. Page 7 Bottom.
Point 8: Table 2: The chemical structures of these compounds should be provided somewhere, at least in a supplementary material.
Response 8: They are now provided in Tables S1 and S2, respectively.
Point 9: Table 5: It is not easy to match the scaffold with the fragments. It is better to mark these structures using name or number. How many structures have been generated in this virtual library? More details need to show the results that lead to the selected compounds,
Response 9: A virtual library containing 320 structures of pyrimidine derivatives was generated combining the chemical scaffolds and substituents depicted in table 5, by using a home-made software. The 90-descriptor set and DF were calculated for this library. Based on these parameters, a virtual screening was performed so that those molecular structures with the values of the descriptors and DF within the thresholds, shown in the Table S3, were selected as candidates. This has also been explained in the article. Page 7 Line 196.
Point 10: The unit of micro-molar concentration is shown in a wrong font (?M), in Table 6 and Page 14 Lines 238-242.
Response 10: We would like express our sincere thanks for the careful reading and helpful remarks of our paper. We had modified the revised version guided by the received reviewer comments.

Reviewer 2 Report
In this work, García-García et al. employed an experimental and computational approach to investigate into potential antituberculosis agents. All investigations to the fight against the Mycobacterium tuberculosis complex remains one of the “hot topics”, so the work fits the current trends in science well. Furthermore, the authors have presented very interesting research results and the work has been also well written. The experiments have been done carefully and no obvious errors or omissions could be detected by referee. Nevertheless, the authors should address the following comments concerning the synthesis used, before the recommendation for acceptance can be given:
- The authors have been used the protocols available in the literature to synthesized potential antituberculosis agents. Unfortunately, the authors do not present the results of identification of these compounds. How was the purity of the compounds confirmed? The relevant results should be included in supplementary materials. The authors informed that the data is included in supplementary materials, but it was not attached to the materials for review.
- In table 1 and in some places in the text (e.g. pages 9 or 20), there is information about incorrectly assigned references. Therefore, the footnotes cannot be verified.
Author Response
Response to Reviewer 2 Comments
In this work, García-García et al. employed an experimental and computational approach to investigate into potential antituberculosis agents. All investigations to the fight against the Mycobacterium tuberculosis complex remains one of the “hot topics”, so the work fits the current trends in science well. Furthermore, the authors have presented very interesting research results and the work has been also well written. The experiments have been done carefully and no obvious errors or omissions could be detected by referee. Nevertheless, the authors should address the following comments concerning the synthesis used, before the recommendation for acceptance can be given:
We would like express our sincere thanks for the careful reading and helpful remarks of our paper.
Point 1: The authors have been used the protocols available in the literature to synthesized potential antituberculosis agents. Unfortunately, the authors do not present the results of identification of these compounds. How was the purity of the compounds confirmed? The relevant results should be included in supplementary materials. The authors informed that the data is included in supplementary materials, but it was not attached to the materials for review.
Response 1: The purity of the synthesized compounds is mainly evaluated by elemental analysis. Representative spectra will be included in the Supplementary Information. Thank you for the observation.
Point 2: In table 1 and in some places in the text (e.g. pages 9 or 20), there is information about incorrectly assigned references. Therefore, the footnotes cannot be verified.
Response 2: Thank you for the information. It is curious. These errors appear in the pdf version, but not in the Word one. We will be alert when reviewing the galley proofs.
Reviewer 3 Report
In this paper, the authors proposed similarity-based virtual screening to find antituberculosis agents based on novel scaffolds. The research is with soundness and novelty. However, there are some problems that should be pointed out.
1. Some references in this paper appear as "Error! Reference source not found". The authors should check and rectify these incorrect references carefully.
2. The "table" and "figure", while being mentioned in the article, should be capitalized. For instance, "... in table 1" should be "... in Table 1".
3. Some contents in Section "2. Results and Discussion" should be written in Section "3. Materials and Methods". The authors should explain the virtual screening procedure in Section "Materials and Methods".
4. The authors should draw a figure or flowchart to illustrate the proposed virtual screening procedure.
5. The authors adopted Linear Discriminant Analysis (LDA) for virtual screening to find antituberculosis agents. The authors should explain more details about the LDA algorithm and the LDA-based virtual screening procedure, to illustrate the proposed method more clearly.
Author Response
Response to Reviewer 3 Comments
In this paper, the authors proposed similarity-based virtual screening to find antituberculosis agents based on novel scaffolds. The research is with soundness and novelty. However, there are some problems that should be pointed out.
We would like express our sincere thanks for the careful reading and helpful remarks of our paper.
Point 1: Some references in this paper appear as "Error! Reference source not found". The authors should check and rectify these incorrect references carefully.
Response 1: Thanks for the info. These errors appear in the pdf version, but not in the Word version. We will be attentive when reviewing the galley proofs.
Point 2: The "table" and "figure", while being mentioned in the article, should be capitalized. For instance, "... in table 1" should be "... in Table 1".
Response 2: Thank you for the observation and sorry for the carelessness. Now, it has been corrected.
Point 3: Some contents in Section "2. Results and Discussion" should be written in Section "3. Materials and Methods". The authors should explain the virtual screening procedure in Section "Materials and Methods".
Response 3: The reviewer may be right, but the results are difficult to understand if the procedure is not previously explained. For this reason, and in order not to make reading the article tedious, we prefer not to repeat the procedure in Materials and Methods, and dedicate this section solely to laboratory tests. In any case, if the reviewer considers it necessary to rearrange the text, we will gladly do so. The virtual screening procedure has been explained in the new version.
Point 4: The authors should draw a figure or flowchart to illustrate the proposed virtual screening procedure.
Response 4: The flowchart has been added. Page 8. Thanks for the suggestion.
Point 5: The authors adopted Linear Discriminant Analysis (LDA) for virtual screening to find antituberculosis agents. The authors should explain more details about the LDA algorithm and the LDA-based virtual screening procedure, to illustrate the proposed method more clearly.
Response 5: This explanation has been introduced in the text. Page 4 Line 98.

Round 2
Reviewer 1 Report
Please see the attached Word file “reviewer1_report2.docx”.

Reviewer 3 Report
The paper has been sufficiently improved. I think this paper can be accepted in present form.Author Response
Thank you very much.
Round 3
Reviewer 1 Report
Author did not respond to the important problems mentioned last time. especially the following problems. These problems require the authors to redesign the whole work plan, not just modify the statements.
1. The computational part is weak. The author just used the discriminant function which only distinguished actives from inactives. Ths method is much lower than QSAR method which gives the ranking of activity. Even QSAR is not sufficient as computational technique, thus more computational treatments should be used to help identify actives. In this manuscript, the author refused to provide the DF values of the top 32 structures, i.e. top 10% of the 320 structures.
2. The best suggested structures 7 and 11b have MIC50 of 81.5 and 80.0 μM (Table 7), while the compounds reported in literature have much smaller MIC50, 0.2~19.6 μM (Table 6). This means that this work does not present any improvement for the rational design of antituberculosis agents.
